# CLIPCleaner: Cleaning Noisy Labels with CLIP

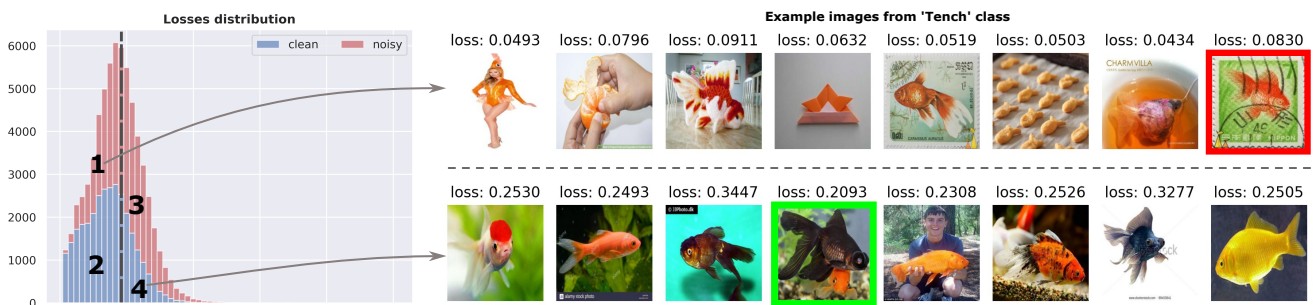

**Figure 1: Loss distribution of the WebVision dataset after one epoch of warmup training with the entire dataset is shown here. Samples are categorized as 'clean' or 'noisy' based on CLIPCleaner's identification. The vertical gray line represents the sample selection boundary imposed by the 'small-loss' mechanism (further details in introduction part). We visualize examples in part 1 and part 4. These examples represent samples classified as 'clean' by 'small-loss' but rejected by CLIPCleaner, and vice versa. Specifically, two samples from the 'Tench' class are highlighted in red and green. We can not find the specific source of red image, but highly-related images can be found with keyword: `1966 Japanese Goldfish Stamp Postage`, while the green one can be originated back to: `https://acnl.fandom.com/wiki/Pop-Eyed_Goldfish`. The red sample is a tench fish postage stamp, semantically similar to real tench images, resulting in a lower loss. The green sample, however, is a photo of a black pop-eyed goldfish, which deviates more from the typical golden tench fish visually.**

## ABSTRACT

Noisy labels pose a significant challenge for machine learning models. Existing sample selection methods for Learning with Noisy Labels (LNL), often based on a strategy like selecting samples with 'small loss', can suffer from 'self-confirmation bias'. This bias arises because these methods rely on the in-training model, which itself might be misled by the noisy labels. Furthermore, solely relying on visual information can introduce biases and challenges like 'hard noise', where noisy labels incorrectly assign samples to semantically similar categories. This paper proposes addressing these challenges by leveraging CLIP, a powerful vision-language model, for sample selection. We introduce CLIPCleaner, which utilizes CLIP's pre-trained zero-shot classifier along with a classifier based on CLIP's vision encoder and the noisy labels themselves. Our approach enables effective offline sample selection. We also provide theoretical justifications and empirical evidence to demonstrate the advantages of CLIP compared to conventional pre-trained models. Compared to current methods that combine iterative sample selection with various techniques, CLIPCleaner offers a streamlined approach while achieving competitive or superior performance on benchmark datasets. Our work highlights the potential of large-scale vision-language models for tackling LNL problems.

## CCS CONCEPTS

• **Computing methodologies** → **Supervised learning**; *Computer vision representations*; Learning under covariate shift.

## KEYWORDS

Sample selection, Noisy Labels, CLIP

## 1 INTRODUCTION

Over the past two decades, deep neural networks have demonstrated exceptional success in various vision tasks, attributed to the existence of high-precision, large-scale datasets such as ImageNet-1K. However, collecting high-quality labels for such datasets is generally time-consuming and labor-intensive. To mitigate the cost, an alternative is automatic labeling (e.g. "webly-labeled" dataset by web-crawling the images and labels). While reducing the cost of manual labeling, it inevitably leads to low-quality noisy labels.

To address the problem of label noise, a variety of methods have been proposed. Some methods, aim to develop robust loss functions [8, 11, 29, 41, 43, 54, 60, 64] or noise transition matrix [12, 15, 26, 34, 48, 53]. However, in practice, these methods are often sub-optimal dealing with high noise ratio and complicated noise.

More recently, methods based on sample selection [19–21, 32, 33, 39, 42, 45, 58] to filter out samples with noisy labels become perhaps the dominant paradigm. For example, the most common sample selection strategy is the 'small-loss' mechanism motivated by the

*ACM MM, 2024, Melbourne, Australia*

© 2024 Copyright held by the owner/author(s). Publication rights licensed to ACM.
ACM ISBN 978-x-xxxx-xxxx-x/YY/MM
https://doi.org/10.1145/nnnnnnn.nnnnnnn

memorization effect, that is, the model tends to fit clean samples earlier than noisy samples in the training process thus resulting in relatively smaller losses for the clean ones. Following this, most of methods focus primarily on improving sample selection mechanisms, including different variants of 'small-loss' strategy [1, 23, 49], and utilizing kNN [2, 7, 31] or graph models [46, 47] based on samples' feature space for sample selection. However, these methods are inherently affected by the label noise as they still rely on the current in-training model, leading to the infamous self-confirmation bias. Some methods [13, 56] attempt to alleviate self-confirmation bias through model co-training, but this approach noticeably introduces additional computational overhead. Moreover, these methods solely rely on the visual information within the images, which can readily lead to biased sample selection outcomes, as exemplified in cases of 'hard noise' - noisy sample exhibits a highly visual similarity with its incorrectly labeled class, as illustrated in fig. 1.

To address the aforementioned issues, this paper proposes utilizing popular vision-language model - CLIP [35], for sample selection. Like any pre-trained models, CLIP is unaffected by the label noise in the collected dataset thus avoiding 'self-confirmation' bias. More importantly, CLIP's distinctive language modality and zero-shot classifier allow us to compensate for the biases that may arise from solely relying on visual information for sample selection. For instance, this allows us to identify 'hard noise' (fig. 1) that is difficult to distinguish using only the vision modality.

**To the best of our knowledge, we are the first to employ a large-scale vision-language model, particularly leveraging its language modality, for sample selection.** Specifically, we simultaneously utilize CLIP's zero-shot classifier and an easily-inducible classifier based on noisy labels and CLIP's vision encoder. We name this method *CLIPCleaner* and theoretically and empirically demonstrate its effectiveness and unique advantages. To ensure the efficiency of the method and facilitate seamless comparison with existing methods, unlike common transfer learning techniques such as model fine-tuning [9], knowledge distillation [44], and prompt-based learning [3, 62], *CLIPCleaner* **does not involve training/fine-tuning the CLIP model.** Furthermore, to evaluate the performance of *CLIPCleaner* on existing datasets, **we introduce a minimal semi-supervised learning method tailored for noisy datasets, namely *MixFix*.** In detail, we gradually introduce ignored clean samples and re-label noisy samples to expand the initial clean subset selected by *CLIPCleaner*.

**By leveraging *CLIPCleaner* and *MixFix* we establish a simple two-step framework for LNL**: initiating with sample selection using *CLIPCleaner* and then perform semi-supervised learning solely using *MixFix*. Compared to existing methods involving iterations of sample selection and model training, **our approach features a simpler structure and aligns better with end-to-end training logic** when the noise information in the dataset is unknown. Moreover, *CLIPCleaner* **can serve as a plug-in module for existing methods**, which further shows great potential of CLIP in learning with noisy labels. **Despite its simplicity, our method achieves competitive and superior performance on various datasets**, including CIFAR10/CIFAR100 with synthetic noise (symmetric, asymmetric, and instance-dependent), as well as real-world noisy datasets like Red Mini-ImageNet, WebVision, Clothing1M, and ANIMAL-10N.

## 2 RELATED WORKS

*Sample selection for noisy dataset.* Most of the recent sample selection methods do so, by relying on the in-training model, for example the per-sample losses [1, 13, 18, 23] or model predictions [30, 38, 55]. A few works focus on further improving the sample selection quality by modelling the loss with markov process [49] or dynamically select samples with multiple metrics [63]. In addition to selecting samples based on the model classifier, some works also try to utilize the feature representations for sample selection. Wu et al. [46] and Wu et al. [47] try to build a kNN graph and identify clean samples through connected sub-graphs, while Feng et al. [7] and Ortego et al. [31] propose to utilize a simpler kNN in feature space to alleviate the effect of noisy labels. Some recent methods involving contrastive learning also identify clean sample pairs based on neighborhood relationships in the feature space [25] or fit Gaussian distributions to model the clean distribution [16]. However, these methods remain unstable and prone to self-confirmation bias, especially in strong noise scenarios, due to its intrinsic reliance on the in-training model based on noisy dataset.

*Utilization of auxiliary model.* To alleviate self-confirmation bias, the utilization of an auxiliary noise-free model is reasonable and straightforward. Related to us, some methods also try to use pretrained noise-free models for learning with noisy labels. Cheng et al. [6], Zheltonozhskii et al. [61] propose to utilize self-supervised pretraining since it can learn good representations in the label-free case. Bahri et al. [2] utilize the pre-logit space of the pretrained model along with the kNN classifier for sample selection. Zhu et al. [65] follow the same idea and also involve CLIP, but they only utilize its vision encoder as a common pretrained encoder without utilizing the language encoder. We emphasize that language modality is critical as a supplementary modality.

## 3 METHOD

In section 3.1, we cast the learning with noisy labels problem in a formulation that covers mainstream sample selection methods. We also provide essential details about the CLIP model. In section 3.2, we elaborate our sample selection method, namely *CLIPCleaner*. In section 3.3, we introduce our semi-supervised learning method, namely *MixFix*. In section 3.4 , we theoretically analyze the unique advantage of using CLIP for sample selection over common pretrained models. In section 3.5, we provide further discussions on the topics of sample selection and the use of the CLIP model for LNL.

### 3.1 Preliminary

*Sample selection with noisy labels.* Given a dataset of training samples $(x_i, y_i)_{i=1}^N$ *i.i.d* sampled from a noisy joint distribution $P(x, y)$ with support as $\sup(P) = \{x \in \mathbb{R}^{C \times H \times W}, y \in \{1, ..., K\}\}$ where $K$ denotes the number of semantic classes, the goal of our method is to learn a classifier $f$ that can accurately predict the true labels $y$ for new, unseen examples. Let us denote the clean joint distribution as $P^{true}(x, y)$. Most sample selection methods aim to approximate and optimize the unbiased empirical risk of $f$ on the clean joint distribution $P^{true}(x, y)$ with samples from noisy joint distribution $P(x, y)$: $\hat{R}^{true}(f) = \frac{1}{N} \sum_{i=1}^N w_i L(x_i, y_i; f)$, where $w_i$ are the sample weights.

Particularly, with optimal weights ($w_i = P^{true}(y_i|\boldsymbol{x}_i)/P(y_i|\boldsymbol{x}_i)$) we can achieve risk-consistent learning[1]. However, since $P^{true}(y_i|\boldsymbol{x}_i)$ and $P(y_i|\boldsymbol{x}_i)$ are typically both unknown for $\boldsymbol{x}_i$, the objective of sample selection methods often revolves around estimating these two to subsequently estimate the optimal weights. In general, the noisy label $y_i$ can serve as a confident proxy of the noisy distribution $P(y_i|\boldsymbol{x}_i)$, making our focus on utilizing an additional auxiliary classifier $\tilde{P}(y_i|\boldsymbol{x}_i)$ to estimate $P^{true}(y_i|\boldsymbol{x}_i)$. In addition, it is commonly accepted to restrict the weights as binary since for most classification datasets, $P^{true}(y_i|\boldsymbol{x}_i)$ tends to be highly centered around only one class. As a result, the optimal weight $w_i$ usually leans towards either 0 or 1 for most samples. Here, we propose a concise form sufficient to comprehensively represent most existing sample selection methods:

$$\tilde{w}_i = \mathbb{G}(\tilde{P}(y_i|\boldsymbol{x}_i), y_i) \in \{0, 1\}, \tag{1}$$

where $\mathbb{G}$ denotes a specific sample selection mechanism, such as the 'small loss' strategy, to further refining the estimation.

*CLIP.* We briefly introduce the CLIP model [35], which is currently one of the most prevalent vision-language models. CLIP aims to learn from a dataset of image-text pairs, denoted as $(\boldsymbol{x}'_i, \boldsymbol{z}_i)_{i=1}^M$ (we use $x'$ here for the CLIP training images to discriminate from above in-question dataset), which is *i.i.d.* sampled from a hidden joint distribution $Q(\boldsymbol{x}, \boldsymbol{z})$ with support as $\sup(Q) = \{\boldsymbol{x} \in \mathbb{R}^{C \times H \times W}, \boldsymbol{z} \in \mathbb{R}^d\}$. We have below as CLIP training loss:

$$L(\boldsymbol{x}'_i, \boldsymbol{z}_i; g, h) = \frac{1}{2}\Big( -\log \frac{\exp(g(\boldsymbol{x}'_i)^T h(\boldsymbol{z}_i))}{\sum_{j=1}^M \exp(g(\boldsymbol{x}'_i)^T h(\boldsymbol{z}_j))}$$
$$-\log \frac{\exp(g(\boldsymbol{x}'_i)^T h(\boldsymbol{z}_i))}{\sum_{j=1}^M \exp(g(\boldsymbol{x}'_j)^T h(\boldsymbol{z}_i))}\Big). \tag{2}$$

Here, $g$ and $h$ denote the vision and language encoder, respectively. Intuitively, the CLIP model tries to maximize the correspondence between relatedrelated image-text pairs.

## 3.2 CLIPCleaner: sample selection with vision-language models

In this section, we propose a new sample selection method based on CLIP, namely *CLIPCleaner*. According to eq. (1), our method (actually nearly all sample selection methods) is divided into two main steps: *1. estimate $\tilde{P}(y_i|\boldsymbol{x}_i)$; 2. calculate weight $\tilde{w}_i$ with specific $\mathbb{G}$*. To enable the analysis between text $\boldsymbol{z}$, image $\boldsymbol{z}$ and label $y$, we consistent the notations for CLIP's training dataset and the in-question noisy dataset. Specifically, we extend the in-question noisy dataset to be *i.i.d* sampled from $P(\boldsymbol{x}, y, \boldsymbol{z})$ (actually from its marginalization), where $\sup(P) = \{\boldsymbol{x} \in \mathbb{R}^{C \times H \times W}, y \in [0, 1, \ldots, K], \boldsymbol{z} \in \mathbb{R}^d\}$; similarly, we extend the sampling distribution of CLIP's training dataset to $Q(\boldsymbol{x}, y, \boldsymbol{z})$. Here we assume $\sup(P) \subset \sup(Q)$.

*3.2.1 Estimate $\tilde{P}(y_i|\boldsymbol{x}_i)$.* We consider two options for estimation: directly utilizing CLIP's zero-shot classifier, or, ignoring CLIP's language modality and treating its vision encoder as a regular pre-trained model and training a new classifier atop it with in-question noisy dataset.

---

[1]Please refer to Supplementary E for details. We omit the variables for brevity, e.g, $P(y = y_i | \boldsymbol{x} = \boldsymbol{x}_i)$ as $P(y_i|\boldsymbol{x}_i)$.

*Option 1: Estimate $\tilde{P}(y_i|\boldsymbol{x}_i)$ with CLIP zero-shot classifier.* Firstly, we assume the causal mechanism for $P$ and $Q$ as: $\boldsymbol{x} \rightarrow \boldsymbol{z} \rightarrow y$ where $\boldsymbol{z}$ denotes the description text and $y$ denotes the semantic label thus we have $y \perp \boldsymbol{x} \mid \boldsymbol{z}$. Roughly speaking, we assume that the semantic label $y_i$ can be independently generated based on a decent image description $\boldsymbol{z}_i$ alone for each image $\boldsymbol{x}_i$. We thus have:

$$\tilde{P}_{zeroshot}(y_i|\boldsymbol{x}_i) = \int Q(y_i|\boldsymbol{z}_i)Q(\boldsymbol{z}_i|\boldsymbol{x}_i)d\boldsymbol{z}$$
$$\propto \int Q(y_i|\boldsymbol{z}_i)Q(\boldsymbol{z}_i, \boldsymbol{x}_i)d\boldsymbol{z}. \tag{3}$$

Thus, we can estimate $\tilde{P}_{zeroshot}(y_i|\boldsymbol{x}_i)$ with above integral by sampling $\boldsymbol{z}_i$ as long as $Q(\boldsymbol{z}_i, \boldsymbol{x}_i)$ and $Q(y_i|\boldsymbol{z}_i)$ is known. Specifically, according to eq. (2), we show that $Q(\boldsymbol{z}_i, \boldsymbol{x}_i)$ can be estimated by the output similarity ($\exp(g(\boldsymbol{x}_i)^T h(\boldsymbol{z}_i))$) of the CLIP model (see Supplementary E). However, $Q(y_i|\boldsymbol{z}_i)$ remains unknown and cannot be learned during the CLIP training process. Most current studies customarily design a single prompt as follows: 'A photo of class name of $y_i$.', implicitly assuming that:

$$Q(y_i|\boldsymbol{z} = \text{'A photo of class name of } y_i \text{.'}) \approx 1.$$

Here, single prompt actually corresponds to sampling a single $\boldsymbol{z}$. Obviously, it is plausible that with more high-quality samplings of $\boldsymbol{z}_i$ instead of only utilizing one single prompt the estimation would be better. In this work, we propose below template to generate multiple prompts $\{\mathcal{P}_j\}_{j=1}^J$ using class-specific features[2]:

$\mathcal{P}_j = $'A photo of {class name of $y_i$}, which is/has {class-specific feature j of class $y_i$}.'

Then we can simplify eq. (3) with above prompts as below:

$$\tilde{P}_{zeroshot}(y_i|\boldsymbol{x}_i) \propto \sum_{j=1}^J \tilde{Q}(\boldsymbol{z} = \mathcal{P}_j, \boldsymbol{x}_i). \tag{4}$$

*Option 2: Estimate $\tilde{P}(y_i|\boldsymbol{x}_i)$ with CLIP vision encoder and noisy dataset.* By treating the CLIP model as an ordinary large-scale pre-trained model, we can also leverage its vision encoder $g$ solely along with the in-question noisy dataset $(\boldsymbol{x}_i, y_i)_{i=1}^N$ to train a new classifier $f'$ (to discriminate it with the original classifier $f$ in section 3.1) for estimation. With the common cross-entropy loss, it is straightforward that the normalized prediction logits serve as an estimate of $\tilde{P}(y|\boldsymbol{x})$:

$$\tilde{P}_{trained}(y_i|\boldsymbol{x}_i) = \text{softmax}(f'(g(\boldsymbol{x}_i)))_{y_i}. \tag{5}$$

By default, we train a *LogisticRegression* classifier as $f'$ with fixed extracted features and noisy dataset. Empirically, we also consider non-parametric *kNN* in ablations Section 4.2.

*3.2.2 Calculate weight $w_i$.* With $\tilde{P}(y_i|\boldsymbol{x}_i)$ estimated above, we can estimate weight $w_i$ for each sample with any applicable sample selection mechanism $\mathbb{G}$. In this work, we consider two simple and popular mechanisms, named $\mathbb{G}_{loss}$ and $\mathbb{G}_{consistency}$. For $\mathbb{G}_{loss}$, we firstly model the per-sample cross-entropy losses ($\{-\log \tilde{P}(y = y_i|\boldsymbol{x}_i)\}_{i=1}^N$) with GMM and then select samples by thresholding its probability belonging to the smaller component. Due to the possible class imbalances and the various semantic diversity of different classes, slightly different than the common approach utilizing a single GMM, we model the losses of samples from each class by a

---

[2]Please refer to Supplementary B for more details about how to generate prompts.

separate GMM model. Please refer to SUPPLEMENTARY D for specific comparisons on seperate GMM and single GMM.

$$\mathbb{G}_{loss} = \mathbb{1}(\mathbb{P}(-\log \tilde{P}(y = y_i|\boldsymbol{x}_i) \in \mathsf{GMM}_{small}) \geq \theta_{loss}).$$

For $\mathbb{G}_{consistency}$, we calculate a consistency measure (defined as the ratio of the probability of noisy label class to the highest class probability) and select samples with high consistency:

$$\mathbb{G}_{consistency} = \mathbb{1}(\tilde{P}(y = y_i|\boldsymbol{x}_i)/\max_k \tilde{P}(y = k|\boldsymbol{x}_i) \geq \theta_{cons}).$$

## 3.3 MixFix: Efficient semi-supervised training by absorbing and relabelling

To evaluate our method on widely-acknowledged benchmarks, we propose a simple semi-supervised learning method for noisy dataset — namely *MixFix*. Please note, the notations employed in this section are defined independently. Specifically, we denote the selected subset and non-selected subset as $(\mathcal{X}_c, \mathcal{Y}_c)$ and $(\mathcal{X}_n, \mathcal{Y}_n)$. Motivated by pseudo-labelling [22] and FixMatch [37], we then inspect each sample's current prediction $\boldsymbol{p}_i$ in non-selected subset with:

$$(w_i, y_i) = \begin{cases} (0, y_i), & \text{if } p_m < \theta_r \text{ and } p_m < \theta'_r \text{ *Drop*} \\ (1, y_i), & \text{if } p_m > \theta_r \text{ and } y_i = y_m \text{ *Absorb*} \\ (1, y_m), & \text{if } p_m > \theta'_r \text{ and } y_i \neq y_m \text{ *Relabel*} \end{cases} \quad (6)$$

Here we denote as $p_m \triangleq \max_l \boldsymbol{p}_i(l)$ and $y_m \triangleq \arg\max_l \boldsymbol{p}_i(l)$. Intuitively, we 'absorb' ignored clean samples ($y_i = y_m$) and 'relabel' noisy samples ($y_i \neq y_m$) with different thresholds in non-selected subset, and progressively append it to initial selected subset to form a dynamic larger training set. Different from existing semi-supervised learning techniques, we typically set $\theta_r \leq \theta'_r$. This helps us make full use of noisy labels to differentiate the 'absorb' and 'relabel' process. To further counter the class imbalance in this new training set, the minority class is over-sampled. Then, we apply a common cross-entropy loss for training with this expanded and class-balanced training set, along with Mixup interpolation [57]. The detailed process is presented in Algorithm 1.

Please note, with selected subset only, *CLIPCleaner* can also be utilized along with existing methods - see SUPPLEMENTARY C for more results. The rationale behind formulating the *MixFix* method, tailored explicitly for noisy datasets, stems from our belief that in scenarios where noise information remains unknown, an end-to-end learning approach is not only more efficient but also stands out as an intuitive and primary choice. This is in contrast to the common style of iterative sample selection and model training.

---

**Algorithm 1: MixFix**.

**Input:** Selected subset $(\mathcal{X}_c, \mathcal{Y}_c)$, non-selected subset $(\mathcal{X}_n, \mathcal{Y}_n)$, $\theta_r, \theta'_r$, max epochs $T$

**while** $i < T$ **do**
  Generate $(\mathcal{X}_r^i, \mathcal{Y}_r^i)$ with eq. (6) ;
  Generate $(\mathcal{X}_t^i, \mathcal{Y}_t^i)$ with $(\mathcal{X}_r^i, \mathcal{Y}_r^i)$ and $(\mathcal{X}_c, \mathcal{Y}_c)$ ;
  Minority over-sampling with $(\mathcal{X}_t^i, \mathcal{Y}_t^i)$ ;
  Model training with $(\mathcal{X}_t^i, \mathcal{Y}_t^i)$ and MIXUP.
**end**

---

## 3.4 Theoretical justification of *CLIPCleaner*

Considering above proposed two options for *CLIPCleaner*, an immediate question is: how does the zero-shot classifier (eq. (4)) compare to the trained classifier (eq. (5)) in estimating $\tilde{P}(y|x)$. If the latter demonstrates comparable or even superior performance to the former, there may be little incentive to employ the CLIP model for sample selection. Rather, pursuing further enhancements to existing large-scale visual-only pre-trained models may yield greater potential. To this end, we conduct a theoretical analysis and compare the distances between the estimated $\tilde{P}(y_i|\boldsymbol{x}_i)$ and true $P^{true}(y_i|\boldsymbol{x}_i)$ of the two options. Specifically, following previous notations, we have below theorems:

THEOREM 3.1 (ESTIMATION WITH ZERO-SHOT CLASSIFIER). *Let $\mathcal{G}, \mathcal{H}$ be the hypothesis space of vision encoder $g$ and language encoder $h$. Let us denote the rademacher complexity as $\mathfrak{R}(\mathcal{G} \circ \mathcal{H})$ of the combined CLIP model. Supposing the range of $L$ from eq. (2) as $[0, l_\infty^{clip}]$ for all $(\boldsymbol{x}, \boldsymbol{z})$ in $\sup(Q)$. Then, for any $\delta > 0$, with probability at least $1 - \delta$ we have the following holds:*

$$d(\tilde{P}_{zeroshot}(y_i|\boldsymbol{x}_i), P^{true}(y_i|\boldsymbol{x}_i)) \leq \varepsilon_{domain}$$

$$+\Delta(\lambda_1\mathfrak{R}(\mathcal{G} \circ \mathcal{H}) + \lambda_2 l_\infty^{clip}\sqrt{\frac{\log 1/\delta}{M}} + \lambda_3\varepsilon_n)$$

*with $\lambda_1, \lambda_2, \lambda_3 > 0$. Here, $\varepsilon_{domain}$ denotes the bias term induced by the domain gap between $Q$ and $P^{true}$, and $\Delta \geq 1$ denotes the bias coefficient induced in designing prompts and sampling in eq. (3).*

THEOREM 3.2 (ESTIMATION WITH TRAINED CLASSIFIER). *Let $\mathcal{F}$ be the hypothesis space of trained classifier $f'$. Let us denote the rademacher complexity as $\mathfrak{R}(\mathcal{F})$ of the trained classifier. Supposing the range of $L$ for training $f'$ as $[0, l_\infty^{noisy}]$ for all $(\boldsymbol{x}, y)$ in $\sup(P)$. Then, for any $\delta > 0$, with probability at least $1 - \delta$ we have the following holds:*

$$d(\tilde{P}_{trained}(y_i|\boldsymbol{x}_i), P^{true}(y_i|\boldsymbol{x}_i)) \leq \varepsilon_{noise}$$

$$+ \lambda_1\mathfrak{R}(\mathcal{F}) + \lambda_2 l_\infty^{noisy}\sqrt{\frac{\log 1/\delta}{N}}$$

*with $\lambda_1, \lambda_2 > 0$. Here, $\varepsilon_{noise}$ denotes the difference term induced by the distribution difference between $P$ and $P^{true}$.*

Please refer to SUPPLEMENTARY F for full derivation. With theorem 3.1 and theorem 3.2, ignoring the uncontrollable and common optimization bound error terms (marked in gray), we confirm that the zero-shot classifier estimation is highly related to domain gap and prompts quality while the trained classifier estimation is affected by the noise of in-question dataset, which is intuitively consistent with our expectation. We also empirically verify that the higher the noise ratio, the greater the performance advantage of zero-shot classifier over the trained classifier (section 4.2). More importantly, $\varepsilon_{noise}$ is always inevitable while $\Delta$ can be easily improved with better prompt engineering and $\varepsilon_{domain}$ can be also reduced by training CLIP with more abundant dataset and thus minimizing the domain gap.

## 3.5 Additional discussion

*To be greedy or conservative?* For all sample selection methods, an inevitable challenge is how to balance the precision and recall of

sample selection. In this paper, we introduce two different classifiers for estimation and two distinct sample selection strategies. Theoretical analysis and subsequent experiments indicate that these different classifiers and selection strategies exhibit their own preferences. In this study, we adopt a conservative sample selection strategy by taking the intersection of different sample selection outcomes, prioritizing the precision of sample selection. Compared to more greedy sample selection strategies, we lean towards relying on the semi-supervised learning strategy - *MixFix* to gradually introduce more samples into training. This can avoid magnifying the influence of noisy samples due to excessively greedy sample selection, but it also has obvious weaknesses, that some 'hard' clean samples will inevitably be missed. We leave the exploration of the optimal sample selection strategy to future work.

*To fully explore CLIP?.* The utilization of the CLIP model for learning with noisy labels remains an area that requires further investigation. To ensure a fair comparison with existing work, we adopt standard sample selection paradigm, refraining from training or fine-tuning the CLIP model [3, 62]. In fact, the current prominent research directions related to CLIP involve fine-tuning the model, specifically through prompt-based learning. However, as expected, recent work (CoOp) has indicated that direct fine-tuning CLIP with noisy datasets can yield poorer performance compared to the initial zero-shot classifier. Therefore, in addition to sample selection, incorporating established techniques for LNL into prompt-based learning with CLIP may also offer promising directions.

## 4 EXPERIMENTS

### 4.1 Experiment details

*4.1.1 Dataset details.* **CIFAR10** and **CIFAR100** datasets comprise 50,000 images. Following established conventions, we assess our method's performance with two types of artificial noise: "symmetric noise," wherein labels are randomly flipped across all samples using a uniform distribution, and "asymmetric noise," wherein labels of visually similar categories, such as Horse ↔ Deer and Dog ↔ Cat, are randomly interchanged. Moreover, we conduct experiments with various noise levels: 20%, 50%, 80% and 90% symmetric noise, as well as 40% asymmetric noise, adhering to the settings in DivideMix ([23]). For instance-dependent noise, we utilize the label noise file provided by [4].

**Red Mini-ImageNet** dataset [17] is a real-world dataset containing a total of 100 categories. It is an extension of the Mini-Imagenet dataset, where noise is introduced at varying ratios. Specifically, noisy images and their respective labels are obtained by crawling the internet, and these noisy images replace the original images in the Mini-ImageNet dataset, with different noise ratios. To ensure a fair comparison with previous studies [10, 52], the images are resized from their original size of 84×84 pixels to 32×32 pixels. Moreover, in accordance with the existing literature [10, 52], we utilize noise ratios of 20%, 40%, 60%, and 80%.

**WebVision** [27] is an extensive dataset comprising 1,000 classes of images obtained through web crawling. In line with previous studies [18, 23, 31], we evaluate our methods using the top 50 classes from the Google Subset of WebVision. The estimated noise ratio for this subset is approximately 20%.

**ANIMAL-10N** [38] is a recently introduced real-world noisy dataset comprises 10 classes of animals. The dataset has undergone manual labeling, with an estimated label noise ratio of around 8%. Similar to the CIFAR datasets, ANIMAL-10N consists of 50,000 training images and 10,000 test images.

**Clothing1M** [50] is a large-scale dataset containing 14 classes of clothing images, obtained by crawling online shopping websites. It consists of a substantial collection of 1 million noisy images. The estimated noise ratio for this dataset is approximately 38.5%.

*4.1.2 Implementation details.* We use CLIP model with VIT-B/32 backbone in all experiments except for specific ablations. In all experiments, our default approach is *CLIPCleaner + MixFix* (**Ours**). By default, we train the network with a SGD optimizer with a momentum of 0.9 in all experiments.

For **CIFAR10** and **CIFAR100**, we use a PresActResNet-18 [14] as the backbone in all experiments following previous works. For CIFAR10, we set $\theta_{loss} = 0.5, \theta_{cons} = 0.8$ for *CLIPCleaner* and $\theta_r = 0.8, \theta_r' = 0.9$ for *MixFix*; For CIFAR10, we set $\theta_{loss} = 0.5, \theta_{cons} = 0.8$ for *CLIPCleaner* and $\theta_r = 0.7, \theta_r' = 0.8$ for *MixFix*. We train both networks with for 300 epochs with a weight decay of 5e-4. The initial learning rate is 0.02 and is controlled by a cosine annealing scheduler. The batchsize is fixed as 128.

For **Red Mini-ImageNet**, we also use a PresActResNet-18 [14] as the backbone following previous works [10, 52]. For *CLIPCleaner*, we set $\theta_{loss} = 0.5, \theta_{cons} = 0.8$. For *MixFix*, we set $\theta_r = 0.8, \theta_r' = 0.95$. We train the network for 300 epochs with a weight decay of 5e-4. The initial learning rate is 0.02 and reduced by a factor of 10 after 200 and 250 epochs. The batchsize is fixed as 64.

For **WebVision**, we use a InceptionResNetv2 as the backbone following [23]. For *CLIPCleaner*, we set $\theta_{loss} = 0.5, \theta_{cons} = 1$. For *MixFix*, we set $\theta_r = 0.7, \theta_r' = 1.0$. We train the network for 150 epochs with a weight decay of 1e-4. The initial learning rate is 0.01 and reduced by a factor of 10 after 80 and 120 epochs. The batchsize is fixed as 32.

For **Clothing1M**, we use a ResNet50 as the backbone following [23] with ImageNet pretrained weights. For *CLIPCleaner*, we set $\theta_{loss} = 0, \theta_{cons} = 0.5$. For *MixFix*, we set $\theta_r = 0.7, \theta_r' = 1.0$. We train the network for 150 epochs with a weight decay of 1e-3. The initial learning rate is 0.002 and reduced by a factor of 10 after 50 and 100 epochs. The batchsize is fixed as 32.

For **ANIMAL-10N**, we use a VGG-19 [36] as the backbone with batch-normalization following [38]. For *CLIPCleaner*, we set $\theta_{loss} = 0.5, \theta_{cons} = 0.8$. For *MixFix*, we set $\theta_r = 0.7, \theta_r' = 0.99$. We train the network with SGD optimizer for 300 epochs with a momentum of 0.9 and weight decay of 5e-4. The initial learning rate is 0.02 and reduced by a factor of 10 after 150 and 250 epochs. The batchsize is fixed as 128.

### 4.2 Ablations study

*Hyper-parameters w.r.t MixFix.* In this section, we ablate on the only two hyperparameters of our semi-supervised training strategy *MixFix*: the 'absorb' threshold $\theta_r$ and the 'relabel' threshold $\theta_r'$. Owing to the precision-recall dilemma when doing sample selection, here we also need to weigh the precision and recall when introducing additional training samples. In table 1 we demonstrate that under different noise ratios, a too high or too low threshold

**Table 1: Ablations on *MixFix* with synthetic CIFAR100 noisy dataset. The *top-3* results are bolded.**

| $\theta_r$ | $\theta'_r$ | Noise ratio | | | |
|---|---|---|---|---|---|
| | | 20% | 50% | 80% | 90% |
| | 0.7 | 76.46 | 74.69 | **69.50** | 62.91 |
| 0.7 | 0.8 | **76.63** | **75.23** | **69.72** | **63.11** |
| | 0.9 | **77.06** | **75.17** | 67.76 | 59.17 |
| | 0.7 | 75.49 | 74.30 | 67.95 | **63.29** |
| 0.8 | 0.8 | 76.36 | **74.90** | 68.86 | **63.42** |
| | 0.9 | **76.66** | 74.50 | 67.37 | 58.09 |
| | 0.7 | 74.53 | 73.49 | 68.74 | 62.22 |
| 0.9 | 0.8 | 75.98 | 74.25 | **68.94** | 62.81 |
| | 0.9 | 75.78 | 74.23 | 67.17 | 59.38 |

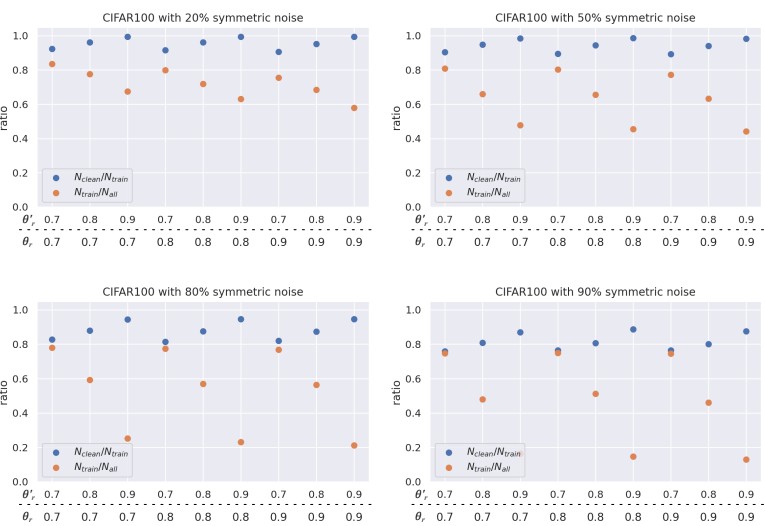

**Figure 2:** $N_{train}$ denotes number of training samples, $N_{clean}$ denotes number of clean training samples and $N_{all}$ denotes number of clean training samples.

**Table 2: Testing accuracy (%) with CLIP zero-shot classifier**

| Model | CIFAR10 | CIFAR100 | Red Mini-ImageNet | WebVision | Clothing1M | ANIMAL-10N |
|---|---|---|---|---|---|---|
| CLIP zero-shot | 89.97 | 63.72 | 78.12 | 73.36 | 39.73 | 76.12 |
| SOTA | 92.68 | 67.7 | 49.55 | 80.9 | 74.84 | 84.6 |
| Ours | **95.15** | **71.17** | **54.21** | **81.56** | **74.87** | **88.14** |

leads to performance degradation, and $\theta_r < \theta'_r$ leads to better performance than setting same value for both thresholds. In fig. 2, we further reveal the inherent mechanism. Especially, after reducing the 'absorb' threshold $\theta'_r$, the proportion of training samples increases and the accuracy of training samples decreases.

*Analyzing CLIP Zero-shot classification as a baseline.* In this section, we consider utilizing CLIP's zero-shot classifier directly on the clean test set, following a procedure that we describe in Section 3.2. In table 2, we present the zero-shot classification results on six common benchmarks and compare them with current SOTA results as well as our own method. It's worth noting that CLIP is utilized with the VIT-B/32 architecture here, while our method and the SOTA methods adopt simpler structures, such as PreResNet-18 for the CIFAR dataset. Therefore, this comparison is indeed 'over stringent'. Even though, we observe that, when compared to directly utilizing CLIP's zero-shot classifier, our method delivers significantly improvements on most datasets and outperforms the SOTA LNL methods on all datasets. We also consider other vision-language models other than CLIP in Supplementary A.

*Analyzing sample selection w.r.t different classifiers and different mechanisms.* In section 3.4, we theoretically conclude that the performance of the zero-shot classifier is influenced by the quality of utilized prompts and the domain gap between CLIP training dataset and the in-question noisy dataset, while the performance of the easily-inducible classifier trained based on CLIP's vision encoder and the in-question noisy dataset is influenced by the noise of the in-question dataset. To validate this, we empirically test with two datasets with controllable noise ratios, that is, the CIFAR10/100 dataset with synthetic noise and the Red Mini-ImageNet dataset with real-world noise.

In fig. 3, we show the sample selection performance and find that: i) As the noise ratio increases, regardless of the dataset, noise types, CLIP backbones or empirical variants of the trained classifier in option 2 (*LogisticRegression* VS *kNN*), the zero-shot classifier (option 1) gradually outperforms the trained classifier. This further validates our theoretical findings in section 3.4; ii) Additionally, we notice that when comparing two different modes for obtaining the training classifier, the *LogisticRegression* classifier empirically exhibits superior performance to the *kNN* classifier. Therefore, we choose the *LogisticRegression* classifier as our default choice for trained classifier; iii) Furthermore, we find that different sample selection mechanisms ($\mathbb{G}_{consistency}$ VS $\mathbb{G}_{loss}$) show distinct advantages and disadvantages on different datasets. Given that noise information is typically unknown in real-world scenarios, as analyzed in section 3.5, we default to a conservative sample selection strategy, which involves utilizing both sample selection strategies and choosing their intersection as final selected subset.

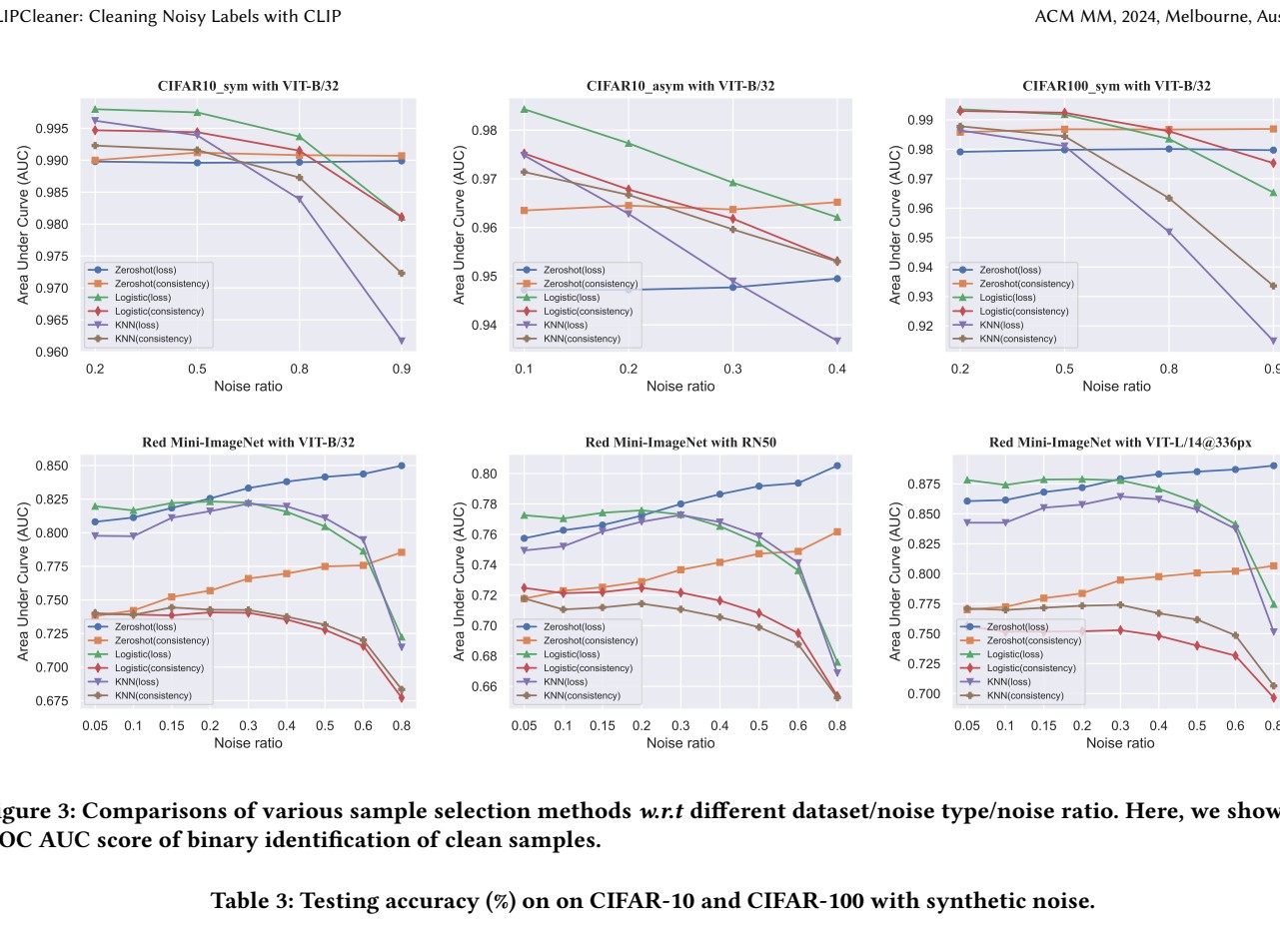

**Figure 3: Comparisons of various sample selection methods *w.r.t* different dataset/noise type/noise ratio. Here, we show the ROC AUC score of binary identification of clean samples.**

**Table 3: Testing accuracy (%) on on CIFAR-10 and CIFAR-100 with synthetic noise.**

| Dataset | CIFAR10 | | | | | CIFAR100 | | | |
|---|---|---|---|---|---|---|---|---|---|
| Noise type | Symmetric | | | | Assymetric | Symmetric | | | |
| Noise ratio | 20% | 50% | 80% | 90% | 40% | 20% | 50% | 80% | 90% |
| CE | 86.8 | 79.4 | 62.9 | 42.7 | 85.0 | 62.0 | 46.7 | 19.9 | 10.1 |
| Co-teaching+ [56] | 89.5 | 85.7 | 67.4 | 47.9 | - | 65.6 | 51.8 | 27.9 | 13.7 |
| F-correction [34] | 86.8 | 79.8 | 63.3 | 42.9 | 87.2 | 61.5 | 46.6 | 19.9 | 10.2 |
| PENCIL [55] | 92.4 | 89.1 | 77.5 | 58.9 | 88.5 | 69.4 | 57.5 | 31.1 | 15.3 |
| LossModelling [1] | 94.0 | 92.0 | 86.8 | 69.1 | 87.4 | 73.9 | 66.1 | 48.2 | 24.3 |
| DivideMix [23] | **96.1** | 94.6 | 93.2 | 76.0 | 93.4 | 77.3 | 74.6 | 60.2 | 31.5 |
| ELR+ [28] | 95.8 | 94.8 | 93.3 | 78.7 | 93.0 | 77.6 | 73.6 | 60.8 | 33.4 |
| MOIT [31] | 93.1 | 90.0 | 79.0 | 69.6 | 92.0 | 73.0 | 64.6 | 46.5 | 36.0 |
| SelCL+ [25] | 95.5 | 93.9 | 89.2 | 81.9 | 93.4 | 76.5 | 72.4 | 59.6 | 48.8 |
| TCL [16] | 95.0 | 93.9 | 92.5 | 89.4 | 92.6 | 78.0 | 73.3 | 65.0 | 54.5 |
| Ours | 95.92±0.15 | **95.67±0.28** | **95.04±0.37** | **94.23±0.54** | **94.89±0.16** | **78.20±0.45** | **75.23±0.29** | **69.72±0.61** | **63.11±0.89** |

## 4.3 Results on synthetic noisy dataset

In this section, we firstly evaluate our method on the CIFAR datasets with synthetic symmetric/asymmetric noise. In table 3, We can see that our method gets competitive and better performance in all experiment settings, especially when the noise ratio is high (63.11% testing accuracy with 90% symmetric noise on CIFAR100 dataset). Also, we would like to emphasize that we keep hyper-parameters fixed for all experiments here as we believe the method robustness in a noise agnostic scenario is critical.

To further validate the performance of our method in handling the 'hard noise', we also conduct experiments on instance-dependent noise in table 5. Different from symmetric or asymmetric noise, instance-dependent noise assumes that semantic-similar samples are more prone to get mislabelled, aligning better with our earlier definition of 'hard noise'. Besides, here we here exclude *MixFix* and employ the selected samples for training with cross-entropy loss solely. This exclusion serves to provide an additional proof of the superior sample selection performance of *CLIPCleaner*.

**Table 4: Testing accuracy (%) on Clothing1M.**

| CE | F-correction [34] | RRL [24] | C2D [61] | DivideMix [23] | ELR+ [28] | SSR+ [7] | TCL [16] | Ours | Ours (Co-training) | CLIPCleaner + DivideMix |
|---|---|---|---|---|---|---|---|---|---|---|
| 69.21 | 69.84 | 74.30 | 74.84 | 74.76 | 74.81 | 74.83 | 74.80 | 73.41±0.65 | 74.01±0.47 | **74.87±0.44** |

**Table 5: Testing accuracy (%) on CIFAR10 with instance-dependent noise.**

| Method | Noise ratio | | | |
|---|---|---|---|---|
| | 10% | 20% | 30% | 40% |
| CE | 91.25 | 86.34 | 80.87 | 75.68 |
| F-correction [34] | 91.06 | 86.35 | 78.87 | 71.12 |
| Co-teaching [13] | 91.22 | 87.28 | 84.33 | 78.72 |
| GCE [60] | 90.97 | 86.44 | 81.54 | 76.71 |
| DAC [40] | 90.94 | 86.16 | 80.88 | 74.80 |
| DMI [51] | 91.26 | 86.57 | 81.98 | 77.81 |
| SEAL [4] | 91.32 | 87.79 | 85.30 | 82.98 |
| CE* | 90.76 | 86.08 | 80.64 | 75.27 |
| CLIPCleaner + CE | **92.33±0.37** | **91.06±0.37** | **89.71±0.37** | **88.26±0.37** |

## 4.4 Results on real-world noisy datasets

Finally, in table 6, table 7, and table 8 we show results on the ANIMAL-10N, Red Mini-ImageNet and WebVision datasets, respectively. In summary, our proposed method demonstrates substantial improvements compared to the current state-of-the-art approaches on both large-scale web-crawled datasets and small-scale human-annotated noisy datasets.

**Table 6: Testing accuracy (%) on on WebVision.**

| Methods | WebVision | | ILSVRC2012 | |
|---|---|---|---|---|
| | Top1 | Top5 | Top1 | Top5 |
| Co-teaching [13] | 63.5 | 85.20 | 61.48 | 84.70 |
| DivideMix [23] | 77.32 | 91.64 | 75.20 | 90.84 |
| ELR+ [28] | 77.78 | 91.68 | 70.29 | 89.76 |
| NGC [47] | 79.16 | 91.84 | 74.44 | 91.04 |
| FaMUS [52] | 79.4 | 92.8 | 77.0 | 92.8 |
| RRL [24] | 76.3 | 91.5 | 73.3 | 91.2 |
| SelCL+ [25] | 79.9 | 92.6 | 76.8 | **93.0** |
| SSR+ [7] | 80.9 | 92.8 | 75.8 | 91.8 |
| TCL [16] | 79.1 | 92.3 | 75.4 | 92.4 |
| Ours | **81.56±0.29** | **93.26±0.65** | **77.80±0.25** | 92.08±0.44 |

We note, that the proposed *CLIPCleaner* can also be used in combination with other schemes. In table 4 we show results on the Clothing1M dataset both with our default setting (*CLIPCleaner + MixFix*) and with it incorporated to two additional schemes: first incorporating our method with co-training, and second replacing *MixFix* with DivideMix [23]. We observe that we obtain results that are superior to the current state-of-the-art. Meanwhile, we would like to note that the majority of existing methods have small differences on the Clothing1M dataset despite the fact that they have large performance differences on other datasets. This suggests that additional training techniques may have a greater impact than

**Table 7: Testing accuracy (%) on on Red Mini-ImageNet.**

| Method | Noise ratio | | | |
|---|---|---|---|---|
| | 20% | 40% | 60% | 80% |
| CE | 47.36 | 42.70 | 37.30 | 29.76 |
| Mixup [57] | 49.10 | 46.40 | 40.58 | 33.58 |
| DivideMix [23] | 50.96 | 46.72 | 43.14 | 34.50 |
| MentorMix [17] | 51.02 | 47.14 | 43.80 | 33.46 |
| FaMUS [52] | 51.42 | 48.06 | 45.10 | 35.50 |
| InstanceGM [10] | 58.38 | 52.24 | 47.96 | 39.62 |
| Ours | **61.44±0.45** | **58.42±0.66** | **53.18±0.47** | **43.82±0.87** |

**Table 8: Testing accuracy (%) on ANIMAL-10N.**

| Method | Accuracy |
|---|---|
| CE | 79.4 |
| SELFIE [38] | 81.8 |
| PLC [59] | 83.4 |
| NCT [5] | 84.1 |
| InstanceGM [10] | 84.6 |
| SSR+ [7] | 88.5 |
| Ours | **88.85±0.61** |

sample selection methods on this specific dataset, possibly due to the fact that the Clothing1M dataset is more fine-grained than other datasets. For such fine-grained noisy datasets, sample selection may not be the optimal strategy, as suggested in Section 3.1, where the basis of sample selection methods relies on highly concentrated conditional probabilities for the samples (eq. (1)).

## 5 CONCLUSION

To mitigate the issues of 'self-confirmation bias' and compensate for visual-only modality in current mainstream sample selection methods, in this paper we propose a method utilizing the large-scale vision-language model CLIP for sample selection, called *CLIP-Cleaner*. We substantiate its effectiveness through both theoretically and empirically. Furthermore, we introduce a straightforward semi-supervised learning method tailored for noisy datasets, called *MixFix*, without the need for intricate off-the-shelf techniques. We emphasize that the exploration of utilizing vision-language models for noisy datasets, such as the potential of existing prompt learning techniques, remains an open direction. Additionally, the possibility of a large domain gap between the CLIP model and the target dataset can influence results, indicating a need for more refined vision-language models. Lastly, our experiments suggest that sample selection methods may not be optimal for fine-grained noisy datasets, which presents itself also as one of our future research directions.

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
