# OpenReview forum: "CLIPCleaner: Cleaning Noisy Labels with CLIP"
_acmmm.org/ACMMM/2024/Conference — MM2024 Poster_

### Official Review · Reviewer_iuwG · 2024-05-06

**Rating:** 4
**Confidence:** 3

**Summary:**

1. This paper proposes addressing these LNL challenges by leveraging CLIP.
2. CLIPCleaner utilizes CLIP’s pre-trained zero-shot classifier along with a classifier based on CLIP’s vision encoder and the noisy labels themselves.

**Strengths:**

1. The paper is the first to employ a large-scale vision-language model, particularly leveraging its language modality, for sample selection.
2. The method achieves competitive and superior performance on various datasets.

**Limitations:**

1. The paper is not well-structured. More illustrative diagrams are needed to simplify the explanation of your method and pipeline.

**Suitability:**

2

---

### Official Review · Reviewer_wUu7 · 2024-05-23

**Rating:** 4
**Confidence:** 2

**Summary:**

The authors leverage the large-scale vision-language model CLIP for sample selection, introducing CLIPCleaner. The proposed CLIPCleaner can serve as a plug-in module to existing methods, consistently delivering performance improvements.

**Strengths:**

- Employing pretrained vision-language models, such as CLIP,  to learn with noisy labels is reasonable and interesting.
- The experiments are conducted on various datasets, and the ablation study is comprehensive.

**Limitations:**

- The lack of description regarding the experimental setting. For instance, the choice of the sample selection method is not specified, as all six methods seem to have their advantages in certain situations as depicted in Figure 3. The paper does not clarify which sample selection method and what hyperparameters are used in the subsequent performance comparison.
- Missing references, such as CoOp on line 487, and SOTA in Table 2.

**Suitability:**

3

---

### Official Review · Reviewer_xjeW · 2024-05-31

**Rating:** 5
**Confidence:** 2

**Summary:**

The paper tackles the challenge of label noise learning by utilizing multimodal CLIP and introducing a novel strategy called MixFix. Traditional methods relying solely on visual information often result in biased sample selection, particularly with 'hard noise'. To address this, the authors propose using CLIP, a vision-language model, for more accurate sample selection. They present CLIPCleaner, which combines CLIP’s pre-trained zero-shot classifier and a classifier based on CLIP’s vision encoder with noisy labels, facilitating effective offline sample selection. MixFix enhances semi-supervised training by incorporating absorption and relabeling techniques. The authors support their approach with both theoretical and empirical evidence, demonstrating the advantages of their method over the others.

**Strengths:**

The paper is well written and filled with extensive experiments and theoretical results, showcasing the authors' rigorous approach. Each chapter is meticulously detailed, ensuring clarity and depth in the presentation of their work. The authors thoroughly explain the results for each table, emphasizing special cases to provide deeper insights and highlight the practical implications of their findings. This careful attention to detail and comprehensive analysis adds significant value to the study, making it a robust and insightful contribution to the field of label noise learning.

**Limitations:**

To compare with CLIP zero-shot in Table 2, it would be better to use the same architecture for consistency. In each table, instead of simply listing SOTA, it would be more informative to specify the methods that achieved those results. Additionally, there are typo errors in lines 541, 286, 275, and 267 that need correction. Another recent work, "Enhancing Noisy Label Learning Via Unsupervised Contrastive Loss with Label Correction Based on Prior Knowledge," also uses CLIP for label noise learning and appears to achieve better results. Comparing the current method with this recent work could provide a more comprehensive evaluation on CLIP-based LNL methods.

**Suitability:**

3

---

### Meta-Review · Area_Chair_n4GH · 2024-07-04

**Recommendation:** Accept (Poster)
**Confidence:** 4

**Metareview:**

The manuscript received three acceptance ratings, indicating that all reviewers agreed the manuscript meets the bar of MM.

There is thus no basis to overturn the consensus. The AC recommends the submission to be accepted.

Please, however, do account for the comments in the final version. Congrats!